# Study of High-Temperature-Induced Morphological and Physiological Changes in Potato Using Nondestructive Plant Phenotyping

**DOI:** 10.3390/plants11243534

**Published:** 2022-12-15

**Authors:** Boris Lazarević, Klaudija Carović-Stanko, Toni Safner, Milan Poljak

**Affiliations:** 1Department of Plant Nutrition, Faculty of Agriculture, University of Zagreb, 10000 Zagreb, Croatia; 2Centre of Excellence for Biodiversity and Molecular Plant Breeding (CroP-BioDiv), 10000 Zagreb, Croatia; 3Department of Seed Science and Technology, Faculty of Agriculture, University of Zagreb, 10000 Zagreb, Croatia; 4Department of Plant Breeding, Genetics and Biometrics, Faculty of Agriculture, University of Zagreb, 10000 Zagreb, Croatia

**Keywords:** high temperatures, multispectral analysis, chlorophyll fluorescence, 3D multispectral scanning, gas exchange

## Abstract

Potato (*Solanum tuberosum* L.) is vulnerable to high temperatures, which are expected to increase in frequency and duration due to climate change. Nondestructive phenotyping techniques represent a promising technology for helping the adaptation of agriculture to climate change. In this study, three potato cultivars (Agria, Bellarosa and Desiree) were grown under four temperature treatments: 20/15 °C (T1), 25/20 °C (T2), 30/25 °C (T3), and 35/30 °C (T4). Multispectral and chlorophyll fluorescence imaging, 3D multispectral scanning, and gas exchange analysis were used to study the effect of moderate heat stress on potato morphology and physiology and select phenotypic traits most responsive to increased temperatures. The most responsive morphological traits to increased temperatures are related to decreased leaf area, which were detected already at T2. Increased temperatures (already T2) also changed leaf spectral characteristics, indicated by increased red, green, and blue reflectance and decreased far-red reflectance and anthocyanin index (ARI). Regarding chlorophyll fluorescence, increasing temperatures (T2) caused an increase in minimal fluorescence of both dark-adapted (F_0_) and light-adapted (F_0_’) plants. Stomatal conductance, transpiration rate, photosynthetic rate, instantaneous water use efficiency (WUE), and intrinsic water use efficiency increased from T1 to T3 and decreased again in T4. Using recursive partitioning analysis, the most responsive potato phenotypic traits to increased temperature were leaf area projected (LAP), ARI, F_0_, and WUE. These traits could be considered marker traits for further studying potato responses to increased temperatures.

## 1. Introduction

Potato (*Solanum tuberosum* L.) represents one of the most important food crops in the world, along with wheat (*Triticum aestivum* L.), maize (*Zea mays* L.), and rice (*Oryza sativa* L.) [1]. Grown worldwide, it achieves relatively high yields [2] and has high nutritive value [3], which makes it an important crop for reaching global food security and eradication of malnutrition, the two main objectives of the FAO’s 2030 Agenda for Sustainable Development [4]. However, global climate change increases the occurrence and intensity of different unfavorable environmental conditions, subjecting crops to various abiotic stresses and significantly reducing yields, and thus represents a major obstacle in achieving the FAO’s 2030 Agenda for Sustainable Development goals. Global climatic data show that the last seven years were the warmest seven years on record. Moreover, the temperature in 2020 was 1.02 °C higher than the baseline 1951–1980 mean, and 2021 tied for the sixth warmest year in a continuing trend [5]. In addition to the further increase in global temperatures, climate change scenarios predict an increase in frequency, intensity, and duration of heatwaves in the future, which will significantly reduce food production and food quality [6]. Potato is highly vulnerable to high temperatures [7], and it is estimated that climate change will cause a reduction in global potato yields by 18–32% [2].

Tuber initiation, induction, set, and tuber bulking are delayed or even inhibited as the temperature increases from 15 to 23 °C [8]. In addition to the direct effect on tuber development, high temperatures also negatively affect metabolic processes. Optimal temperatures for photosynthesis at the potato canopy level are around 24 °C [9]. Higher temperatures reduce the rate of photosynthesis by enhancing leaf senescence, reducing chlorophyll content, decreasing stomatal conductance [10], increasing photorespiration [11], and decreasing photosystem II (PSII) efficiency [12]. High temperatures adversely affect different plant organs and change plant growth rates and morphology [8,9]. For example, sprout growth and emergence are significantly affected at temperatures higher than 25 °C [13]. The optimal temperature for root growth is 20 °C, whereas higher temperatures limit the allocation of the assimilates to the roots and reduce root growth [8]. In addition, the highest leaf appearance rate occurs at 28 °C [9], whereas the highest rate and duration of leaf growth is at 25 °C [14]. Stem elongation and lateral stem branching are promoted by higher temperatures, up to 30 °C [15]. Thus, higher temperatures (above 25 °C) promote taller plants with more lateral branches and smaller leaves [8]. All this indicates that potato responds to the temperature on the whole plant level, with multiple morphological and physiological changes. This should be considered in different studies aiming to determine the effect of high temperatures on potato growth, tolerance, physiology, and yields. This was also recognized by Hancock et al. [16], who stated that heat stress tolerance is a multigenic trait, and the understanding of basic physiological, biochemical, and molecular responses to elevated temperatures is essential for breeding programs aiming to develop heat-tolerant potato genotypes [16,17]. Moreover, the same authors emphasized the necessity of studying the effects of moderately increased temperatures and mild temperature stresses, compared to traditional heat stress experiments, which subjected plants to high temperatures, because of their significant effects on yields and frequent occurrence, and thus higher importance in the agricultural context [16]. 

One of the critical steps for adapting agriculture to climate change and new growing conditions will be adopting and implementing new technologies [6,18]. Proximal and remote sensing plant phenotyping techniques based on plant multispectral imaging and chlorophyll fluorescence imaging represent such modern technological solutions for studies of complex plant × environment interactions [19,20,21]. Combining such techniques with gas exchange analysis [19,22] enables a comprehensive study of heat stress and nondestructive quantification of stress-induced morphological, physiological, and biochemical changes in potatoes. In addition to gas exchange analysis, other phenotyping techniques were previously only partly used in studying the effect of high temperatures and heat stress in potatoes. Using spectral reflection analysis and calculated vegetation indices, Romero et al. [19] found a good correlation among water index (WI), normalized differential vegetation index (NDVI), simple ratio (SR), and photochemical reflectance index (PRI) with potato foliar area, total water content, the relative growth rate of tubers, leaf area ratio, and foliar area index. Similarly, Ray et al. [23] found a positive correlation between NDVI and soil-adjusted vegetation index (SAVI) with potato leaf area index (LAI). These authors found that hyperspectral indices were more efficient than the LAI for detecting the differences among potato crops under different irrigation treatments. Prashar et al. [24] used infrared tomography to evaluate stomatal behavior in field potato trials and found a negative correlation between canopy temperature and final tuber yield. Li et al. [25] used unmanned aerial vehicles (UAVs) equipped with red, green, and blue cameras for the estimation of potato emergence, whereas Colwell et al. [26] used a similar technique and developed 3D surface models of the potato canopy, from which data on plant height, canopy ground cover, and canopy volume could be obtained [25]. 

This study aims to evaluate the effect of increased temperatures on potato morphology and physiology using nondestructive plant phenotyping techniques. Specific aims were: (i) to quantify traits obtained by nondestructive phenotyping techniques (morphological, multispectral, chlorophyll fluorescence, and gas exchange traits) in potato under increasing temperatures, (ii) to compare these groups of traits in their efficiency to discriminate among potato plants grown under different temperatures, and (iii) to select phenotypic traits of potato that are most responsive to increased temperatures.

## 2. Results

A list of all measured traits with the description is given in Appendix A. Results of the two-way analysis of variance (ANOVA) testing the effects of cultivars, treatments, and cultivar × treatment interactions are shown in Appendix A, whereas Tukey’s HSD pairwise differences are shown in Appendix A.

### 2.1. Effect of Increasing Temperatures on Potato Morphological Traits (MORPH)

After scanning plants with a 3D multispectral scanner, 3D plant models were built from 3D point clouds. Plant 3D models were then used for the quantification of different morphological parameters. Results for all measured morphological traits are shown in Appendix A, and results for selected morphological traits are shown in Figure 1. All measured morphological traits were significantly affected by treatments, cultivars (except the LANG), and treatment × cultivar interactions (Appendix A).

Average plant height increased with increasing temperatures (from T1 to T4), which was most pronounced for cv. Desiree and least pronounced for cv. Bellarosa (Appendix A, Figure 1b). The opposite was found for traits related to leaf area (TLA, LAP, and LAI), which were significantly decreased by increasing temperatures (from T1 to T4) and were most affected for cv. Beallrosa and least affected for cv. Desiree (Appendix A, Figure 1c). This affected DV (calculated as PH × TLA), which decreased with increasing temperatures (from T1 to T4) in Agria and Bellarosa. In contrast, for Desiree, DV increased from T1 to T3 and then decreased in T4 (Appendix A, Figure 1a). In addition to reducing the leaf size, increased temperatures decreased the average leaf angle (Appendix A, Figure 1c), especially for cv. Desiree. Consequently, reduced leaf size and decreased leaf angles, found under higher temperatures, caused increased light penetration depth (LPD/PH) (Appendix A). 

The recursive partitioning model based on morphological traits (MORPH) correctly assigned 84.6% of the plants to their respective temperature treatments. Variables selected by this model were LAP, DV, LANG, and LPD/PH (Figure 2a). This model misclassified six plants (28.6%) from T2 and assigned them as T1, and five plants (23.8%) from T4 were misclassified as T3. Although this model showed the lowest accuracy in the correct assignment of plants into their respective temperature treatments, LAP enabled discrimination between plants from T1 and T2 against T3 and T4 (Figure 2a).

### 2.2. Effect of Increasing Temperatures on Potato Multispectral Traits (MSTs)

Potato multispectral analysis was performed by capturing whole-plant top view images in red (R), green (G), specific green (SpcGrn), blue (B), far-red (FR), and near-infrared (NIR) wavelengths and specific wavelengths for the chlorophyll. These reflectance analyses were used to calculate HUE, SAT, and VAL, and different vegetation indices (NDVI, CHI, ARI, and GLI) (Appendix A).

Temperature treatments significantly affected all measured multispectral traits except SAT (Appendix A). Significant differences were found between cultivars for all measured multispectral traits except HUE, CHI, and ARI. In addition, a significant treatment × cultivar interaction was found for B, NIR, HUE, SAT, GLI, CHI, and ARI (Appendix A).

Increased temperatures (from T1 to T4) generally increased the reflectance in R, G, B, SpcGrn, and FR. For the reflectance in B, the highest increase was found for cv. Agria. The reflectance in NIR was not affected by temperature in Bellarosa, while it increased from T1 to T2 and decreased in T3 and T4 in cv. Agria, and increased from T1 to T3 and then decreased at T4 for cv. Desiree (Appendix A). 

A decrease in most vegetation indices accompanied these changes (increases) in reflection found at higher temperatures. However, a significant decrease in CHI was found only for Desiree in T4 (Appendix A, Figure 3b), whereas HUE, ARI, and NDVI decreased only at T4 for all cultivars (Appendix A, Figure 3c).

Using multispectral traits, recursive partitioning correctly assigned 89.3% of potato plants into their respective temperature treatments. Variables selected by this model were ARI, SpcGrn, and GLI (Figure 2b). This model performed best for assigning plants to T4 treatment (100% of correctly assigned plants). This model could discriminate plants from T4 against plants grown in T1, T2, and T3 already based on ARI (<2.68) (Figure 2b). However, it misclassified two plants from T1 (9.5%), eleven from T2 (52.3%), and five from T3 (23.8%) and assigned them to the wrong group (Figure 2b).

### 2.3. Effect of Increasing Temperatures on Potato Chlorophyll Fluorescence Traits (CFTs) 

Whole-plant chlorophyll fluorescence imaging revealed that temperature treatments affected all measured chlorophyll fluorescence traits. Between studied cultivars, significant differences were found for all CFT except NPQ and qN. In addition, a significant treatment × cultivar interaction was found for all CFT except NPQ and qN (Appendix A). Increasing temperatures increased the minimal, steady state, and maximum fluorescence of both dark- and light-adapted plants; however, a higher increase was found for F_0_ and Fs’ compared to F_m_ and F_m_’, which caused a decrease in F_v_/F_m_ and F_q_’/F_m_’, although only at higher-temperature treatments. Namely, a significant decrease in F_v_/F_m_ was detected only for cv. Agria in T3 and T4 and for Bellarosa in T4, whereas F_q_’/F_m_’ significantly decreased in T3 and T4 for cv. Desiree, and in T4 for cv. Agria and cv. Bellarosa (Appendix A, Figure 4a,b). Similarly, ETR decreased in T4 for cv. Agria and Bellarosa, and in T3 and T4 for Desiree, whereas NPQ, qN, and ɸnpq increased in T4 for all cultivars (Appendix A, Figure 4c,d).

Based on chlorophyll fluorescence traits (CFTs), recursive partitioning correctly assigned 90.5% of the plants to their respective temperature treatments. Variables selected by this model were F_0_, F_m_, qP, and ETR (Figure 5a). This model performed best for assigning plants to T1 treatment (100%). However, it misclassified one plant from T4 (4.7%) and assigned it as T3, two plants from T3 (9.5%) were misclassified as T4, and five plants from T3 (23.8%) were misclassified as T2. Based on F_0_, this model enabled discrimination of plants grown in T4 from T1 and T2 and plants grown in T3 from T1 (Figure 5a).

### 2.4. Effect of Increasing Temperatures on Potato Gas Exchange Traits (GEXTs)

A significant effect of temperature treatments and cultivars was found for all gas exchange traits. In addition, the cultivar × treatment interaction was significant for all gas exchange traits except the Ci (Appendix A). 

Despite a significant cultivar × treatment interaction, generally, stomatal conductance, transpiration rate, and photosynthesis rate increased with increasing temperatures from T1 to T3 and decreased at T4 (Figure 6a–c, Appendix A). In addition, Ci increased for cv. Bellarosa from T1 to T2 and T3, was constant for cv. Agria and cv. Desiree, and decreased significantly for all cultivars in T4 (Figure 6d, Appendix A). WUE decreased from T1 to T3 because increased temperatures caused a higher increase in E compared to A, whereas T4 caused a stronger decrease in A compared to E. For similar reasons (a stronger increase in gs in T2 and T3 compared to A), WUEi decreased from T1 to T3 but then increased again in T4 because of a strong decrease in gs (Appendix A). The opposite behavior found for WUEi was found for the A/Ci ratio (Appendix A).

The highest classification success, 100% correctly assigned potato plants to their respective temperature treatments using recursive partitioning, was achieved using gas exchange traits (Figure 5b). Variables selected by this model were WUE, gs, and E. Using only WUE, the recursive partitioning model enables discrimination of plants from T1 and T2 (WUE < 2.95) against those from T3 and T4 (WUE > 2.95) (Figure 5b).

## 3. Discussion

In this study, three commercial potato cultivars (Agria, Bellarosa, and Desiree) were grown over four different day/night temperature treatments (20/15, 25/20, 30/25, and 35/30 °C), aiming to evaluate the effect of increased temperatures on potato morphology and physiology using nondestructive plant phenotyping techniques. Although the results of this study show a significant effect of the cultivar and treatment x cultivar interaction for most studied traits, a general effect of increased temperatures that alter potato physiological processes and growth could be noticed. Similarly, Tang et al. [27] studied the effect of heat stress on 55 commercial potato cultivars, and despite significant differences among studied cultivars, they found that heat stress generally reduced leaf size, increased plant height and chlorophyll content (SPAD values), and decreased tuber mass. Similar morphological changes were found in this study. Increased temperatures caused an increase in plant height and a decrease in leaf area (LAP, LAI, and TLA). Decreased leaf area and decreased leaf angle increased light penetration depth under higher temperatures. LAI, LAP, LANG, and LPD/PH are traits that were significantly affected already in T2 compared to T1, whereas PH and TLA were affected only at higher temperatures (T3 and T4). The recursive partitioning model based on MORPH showed the lowest accuracy for assigning plants into their respective temperature treatments compared to models that used other groups of traits. The reason for the lower accuracy or higher misclassification, especially among T1 and T2 and among T3 and T4, in this model could be the variability between studied cultivars in their morphological traits and the higher variability in cultivar response to the temperature treatments. However, LAP enabled discrimination between plants from T1 and T2 against T3 and T4, and misclassification occurred only between T1 and T2 and between T3 and T4.

In addition to leaf area reduction, increased temperatures affected leaf spectral properties. With the increasing temperatures from T1 to T4, reflectance in R, G, B, SpcGrn, NIR, and FR significantly increased. Higher reflectance, especially in R and B, indicates lower light absorption, which is often related to the lower chlorophyll content [28]. However, in this study, a significant decrease in CHI and NDVI was found only in the highest-temperature treatment (T4), indicating that the obtained changes in reflectance were caused not only by a decrease in chlorophyll content but also by factors related to leaf morphology, leaf angle, water content, etc. [29]. Previous studies reported inconsistent results about the effect of high temperatures on potato chlorophyll content. Hancock et al. [16] found a significant decrease (up to 20%) in chlorophyll a and b content in the leaves of potato plants grown under 30/20 °C compared to 22/16 °C. In contrast, Tang et al. [27] found increased chlorophyll content (SPAD values) in potato plants under the heat stress (35/28 °C). In our study, the average value of the chlorophyll was estimated using the top view of the plant image , which enabled the estimation of chlorophyll content at the whole-plant level, compared to the abovementioned studies that measured chlorophyll content on the limited area of the single leaf. Compared to CHI that significantly decreased only in plants grown under T4, anthocyanin content (ARI) decreased already at T2. Moreover, ARI was selected by recursive partitioning based on MST as the variable that enables discrimination between plants grown in T4 from plants grown in T1, T2, and T3. It is known that high temperatures strongly affect anthocyanin synthesis through the degradation of the regulator of anthocyanin biosynthesis HY5 [30]. 

Regarding CFT, increasing temperatures had the most profound effect on the minimal fluorescence of both dark-adapted (F_0_) and light-adapted (F_0_’) plants, which increased already in T2. This rise in minimal fluorescence indicates the inactivation of PSII reaction centers [31]. However, due to the increase in F_m_, there was a less pronounced effect on the reduction in F_v_/F_m_. Moreover, although F_v_/F_m_ significantly decreased in T3 and T4, values higher than >0.8 indicate a good fitness of all plants in terms of PSII [31,32]. As with multispectral traits, this may be explained by the changed optical properties of the leaf [31]. Light-adapted chlorophyll fluorescence parameters significantly decreased only in T4 (at 35 °C). This was indicated by the decrease in the proportion of open PSII centers (qP), a concomitant decrease in energy used in photochemistry (F_q_’/F_m_’ and ETR), and an increase in thermal dissipation (NPQ, qN, and ɸnpq). It was previously shown that the transfer of potato leaves from 25 °C to temperatures between 30 and 35 °C does not negatively affect PSII activity, whereas temperatures above 38 °C caused an inhibition in the PSII function [33]. Based on chlorophyll fluorescence traits, recursive partitioning was best for assigning plants to T1 treatment (100%) but could not discriminate between plants grown in T2 and T3 and between plants grown in T3 and T4. However, based on F_0_, this model enabled the discrimination of plants grown in T4 from T1 and T2.

Gas exchange analysis showed a significant increase in stomatal conductance, transpiration, and photosynthetic rate with the increasing temperature from T1 to T3. The highest-temperature treatment (T4) decreased the gs to levels such as those found in T1. This drop in gs is probably related to an increased water vapor pressure deficit (WVPD), which increased up to 1.95 kPa under T4. The photosynthetic rate decreased at T4 due to stomatal limitation. However, increased temperatures had a stronger effect on E and gs than on A, and thus, WUE and WUEi decreased with increasing temperatures from T1 to T3, and then WUEi increased again at T4. The fact that increased temperatures had a stronger effect on gs and E compared to A was further confirmed by the recursive partitioning model, which correctly assigned plants to the respective temperature treatment groups based on WUE, gs, and E. Using only WUE, this model enabled discrimination of plants from T1 and T2 against those from T3 and T4. Recursive partitioning based on gas exchange traits achieved the best assignment success. This result is probably caused by the gas exchange measurement technique, which was always performed on a single leaf of the same size and development stage, compared to the whole-plant imaging and scanning, which captured the leaf/plant spatial variability. 

Similarly, increased gs and A under well-watered conditions in potatoes acclimated to moderately high temperatures (up to 30 °C) were obtained by Hancock et al. [16]. These authors found an enhanced expression for photosynthesis-related genes (associated with PSII, RuBisCo, RuBisCo activase, etc.) in potato plants acclimated to moderately high temperatures. Although optimal temperatures for photosynthesis at the potato canopy level are around 24 °C [9], plants tend to acclimate to their growth temperatures, and photosynthetic acclimation to high temperature shifts the optimum temperature for photosynthesis [11]. Thus, the increased gs, E, and A found with the increased temperature treatments (T1–T3) indicate this heat acclimation process. 

One of the aims was to select phenotypic traits most responsive to increased temperatures using recursive partitioning models. The recursive partitioning for all groups of traits did not include the effect of the cultivar. Thus, it can be assumed that the traits selected by these models are the ones that had the most uniform response to increased temperature over the tested cultivars. Because the studied cultivars showed similar reactions to the temperature treatments, and although these results were similar to those previously described [7,8,9,16,27], this limited variability in studied traits could increase the accuracy of the used models and affect the phenotypic traits that were selected for the assignment of plants into their respective treatments.

Thus, these results should be confirmed using a higher number of cultivars and include cultivars with different responses to heat stress (heat-tolerant vs. heat-sensitive). Moreover, this study was performed on young potato plants, and measurements were performed after ten days of acclimation, which could also affect the traits selected by recursive partitioning models. In addition to the genotypic variability, the magnitude of the stress response depends on the severity, duration, and the plants’ growth stage [34],. Young potato plants are most sensitive to heat stress [35], whereas the acclimation period enables a shift from plant growth to protective mechanisms [16,36]. Thus, studying the effect of increased temperatures on selected phenotypic traits after a prolonged acclimation period, in different potato developmental phases, especially under field conditions, would be of high interest from an agricultural point of view.

## 4. Materials and Methods

### 4.1. Experimental Setup and Growth Conditions

The experiment was conducted in growth chambers 2 × 3 × 2.5 m (width × length × height) (Kambic laboratory equipment d.o.o., Metlika, Slovenia, EU) located at the Faculty of Agriculture, Zagreb, Croatia. Three potato cultivars, cv. Agria, cv. Bellarosa, and cv. Desiree, were planted in plastic tubes (50 cm height, 15 cm diameter) filled with 2.5 kg of Potting Substrate (Klasmann-Deilmann GmbH, Geeste, Germany, EU). One tuber (average fresh weight 50 g) per tube was planted at 7 cm depth. Emergency and early growths were conducted at 16/8 h photoperiod, 20/15 °C thermoperiod, 65% relative air humidity, and 250 µmol PAR m^−2^ s^−1^ provided by light-emitting-diode (LED) lights PhenoLight3 (PhenoVation, Wageningen, The Netherlands, EU). Light intensity was determined at the upper side of the tube (at substrate level) using an LI-180 spectrometer (LI-COR Biosciences, Lincoln, Nebraska, USA). Plants were regularly irrigated with distilled water, keeping the substrate volumetric water content (VWC) at 35% throughout the experiment. The substrate VWC was regularly monitored using a substrate-calibrated WET Sensor and HH2 Moisture Meter (Delta-T Devices Ltd. Cambridge, United Kingdom). After emergence, all plants were thinned to the single main stem. Two weeks after the emergence, seven plants per cultivar (a total of 84 plants) were subjected to four different temperature treatments (thermoperiods): 20/15 °C (T1), 25/20 °C (T2), 30/25 °C (T3), and 35/30 °C (T4). Temperature treatments lasted for ten days, during which all other environmental factors were kept the same as during the emergency and initial growth phases.

All measurements were performed ten (10) days after the onset of temperature treatments. A list of all measured traits, along with equations, is shown in Appendix A.

### 4.2. Morphological Measurements

Morphological traits were measured nondestructively using the PlantEye F500 multispectral 3D scanner (Phenospex, Heerlen, The Netherlands). A detailed description of the PlantEye F500, along with the resolution and the analyzed wavelengths, is given in Lazarević et al. [37]. In brief, after scanning the experimental plants, Phena software (Phenospex, Heerlen, The Netherlands) was used to build the 3D plant model from the 3D point cloud. Examples of generated 3D plant models are given in Figure 7. From the 3D plant model, morphological traits were calculated using HortControl software (Phenospex, Heerlen, The Netherlands). Analyzed morphological traits were: plant height (PH; mm), leaf area projected (LAP; cm^2^), total leaf area (TLA; cm^2^), digital volume (DV; cm^3^), leaf area index (LAI, mm^2^ mm^−2^), leaf angle (LANG; °), and light penetration depth (LPD; mm). Due to changes in plant height under different temperature treatments, which also affected light penetration depth, the LPD parameter was replaced by a parameter calculated as LPD/PH.

### 4.3. Chlorophyll Fluorescence Measurements

Chlorophyll fluorescence imaging was performed on the whole plant using CropReporter^TM^ (PhenoVation B.V., Wageningen, The Netherlands). The CropReporter^TM^ is located in the room next to the growth chambers. Plants were manually transferred to the CropReporter^TM^ room and were left for dark adaptation for 3 h. During the period of dark adaptation and subsequent measurements, the imaging room temperature was kept constant at 23 °C. Thus, plants were not subjected to temperature treatments during this period. A detailed description of the CropReporterTM and the protocol used for chlorophyll fluorescence imaging is given in Lazarević et al. [37]. Briefly, plants were imaged from the 70 cm distance using the optimized quenching protocol [38]. For the fluorescence excitation, a saturating light pulse (4500 μmol m^−2^ s^−1^, for 800 ms) was used. Minimum chlorophyll fluorescence (F_0_) was measured at 10 μs and maximum chlorophyll fluorescence (F_m_) was measured after saturation (800 ms). After these measurements, plants were relaxed for 15 s in the dark and were then light-adapted for five minutes under the 250 μmol m^−2^ s^−1^ (PPFD). After light adaptation, a saturation pulse (4500 μmol m^−2^ s^−1^ for 800 ms) was applied again. Steady-state fluorescence (Fs’) was measured at the onset of the saturating pulse and maximum chlorophyll fluorescence (F_m_’) was measured at saturation. After the measurement, the actinic light was turned off, and in the presence of far-red light, the minimal fluorescence yield of the illuminated plant (F_0_’) was estimated. Collected images were processed by DA^TM^ software (PhenoVation B.V., Wageningen, The Netherlands). From the measured F_0_, F_m_, F_s_’, F_m_’, and F_0_’, different chlorophyll fluorescence parameters were calculated: Maximum quantum yield of PSII (F_v_/F_m_) = (F_m_ − F_0_)/F_m_ [39].Effective quantum yield of PSII (F_q_’/F_m_’) = (F_m_’ − F_s_’)/F_m_’ [39].Electron transport rate (ETR) = F_q_’/F_m_’ × PPFD × 0.5 × 0.83 [39].Non-photochemical quenching (NPQ) = (F_m_ − F_m_’)/F_m_’ [40].Coefficient of Photochemical Quenching (qP) = (F_m_’ − F_s_)/F_v_ [41].Coefficient of Non-photochemical Quenching (qN) = 1 − (F_m_’ − F_0_’)/(F_m_ − F_0_) [41].Estimation of ‘open’ Reaction Centers based on a Lake Model (qL) = ((F_m_’ − F_s_’) × F_0_’))/((F_m_’ − F_0_’) × F_s_’)) [42].Quantum Yield of Non-regulated Non-Photochemical Energy Loss in PSII (ɸnq) = 1/(NPQ + 1 + qL(F_m_/F_0_ − 1)) [43].Quantum yield of Regulated Non-Photochemical Energy Loss in PSII (ɸnpq) = 1 − ɸpsII − 1/(NPQ + 1 + qL(F_m_/F_0_ − 1)) [43].

Examples of generated chlorophyll fluorescence images are shown in Figure 8.

### 4.4. Multispectral Measurements

Multispectral images were obtained following the chlorophyll fluorescence imaging under the actinic light (250 μmol m^−2^ s^−1^) using CropReporterTM. Reflectance images in different spectral wavelengths, red (R) (640 nm), green (G) (550 nm), specific green (SpcGrn) (510–590 nm), blue (B) (475 nm), far-red (FR) (710 nm), and near-infrared (NIR) (769 nm), and specific wavelengths for the chlorophyll (730 nm), were collected. Information about the absolute spectral reflectance and different vegetation indices was calculated by DATM software. Color analysis was presented as an absolute reflection in R, G, and B and as hue (HUE), saturation (SAT), and value (VAL). HUE was calculated from R, G, and B values and is represented as one channel arranged in a rainbow color chart with values 0–360°. The saturation (SAT) of each color represents its intensity (pale or intense color), and the value (VAL) shows if the color is dark or bright.

In addition to color parameters, different vegetation indices were calculated:Normalized Differential Vegetation Index (NDVI) = (NIR − Red)/(NIR + Red) [44].Chlorophyll index (CHI) = (R730)-1 − (R769)-1 [45].Anthocyanin index (ARI) = (R550)-1 − (R700)-1 [46].Green Leaf Index (GLI) = (2 × Green − Red − Blue)/(2 × Green + Red + Blue) [47].

Examples of generated multispectral images are shown in Figure 9.

### 4.5. Gas Exchange Measurements

Gas exchange measurements were conducted using the portable photosynthesis system LI-6800 (LI-COR Biosciences Inc., Lincoln, NE, USA) at the main leaflet of the youngest fully developed leaf (fourth leaf from the top of the canopy). Measurements were performed using a 3 × 3 cm, 6800-02P clear-top chamber (LI-COR Biosciences Inc., Lincoln, Nebraska, USA). Measurements were recorded under a saturating light of 1200 µmol PAR m^−2^ s^−1^ and 90:10% red-to-blue wavelength ratio, provided by the small light source 6800-02 (LI-COR Biosciences Inc., Lincoln, NE, USA). The cuvette temperature was set to meet the applied daytime treatment temperature regimes (T1–T4): 20, 25, 30, and 35 °C, respectively. The cuvette relative humidity was set at 70%, giving a leaf-to-air vapor pressure difference between 0.85 and 1.95 kPa, T1 to T4, respectively. The concentration of CO_2_ was set at 400 µmol mol^−1^ and the flow rate was set at 600 µmol s^−1^. Recordings were performed after CO_2_ and H_2_O stability criteria were met, and automatic matching was performed after each record.

### 4.6. Statistical Analysis

The analysis of variance (ANOVA) using the general linear model (GLM) was performed in JMP^®^ Pro 16 (SAS Institute Inc., Cary, NC, USA). The model included the fixed effects of treatments (T1–T4), cultivars (Agria, Bellarosa and Desiree), and treatment × cultivar interactions. Individual plants were used as pseudo-replications and were treated as a random factor. In the case of the significant F test, pairwise comparisons were performed using Tukey’s Honest Significant Difference (HSD) post hoc test.

Recursive partitioning was used to identify the traits within each group (MORPH, CFT, MST, and GEXT) that are most responsive to increased temperatures [48]. This method builds classification models using a multi-stage procedure where the resulting models can be represented as decision trees. At each stage, the variable that best separates the data into groups defined by the classification variable (treatment) is selected, and data are subsequently split until the model reaches the best possible classification of data into predefined groups (treatments). The efficiency of the final classification models was estimated by the ratio of correctly classified data and the total number of data samples (84). Recursive partitioning was performed using R package rpart [49].

## 5. Conclusions

Potato is very sensitive to heat, and its acclimation to high temperatures involves complex changes in the physiological and morphological levels. Nondestructive plant phenotyping techniques enable the study of these complex reactions and represent technological solutions that will have a significant role in the adaptation of agriculture to climate change and the new growing conditions they bring. Although these results should be confirmed using a higher number of potato cultivars and including the cultivars with different responses to heat stress (heat-tolerant vs. heat-sensitive), as well as under the field conditions after a prolonged period of adaptation to increased temperatures, we selected the most responsive phenotypic traits (LAP, ARI, F_0_, and WUE) to increased temperatures. These traits could be considered as marker traits for the identification of heat stress in potatoes and in screening for heat tolerance among potato germplasms. This set of traits could also be used to increase the throughput in phenotyping studies.

## Figures and Tables

**Figure 1 plants-11-03534-f001:**
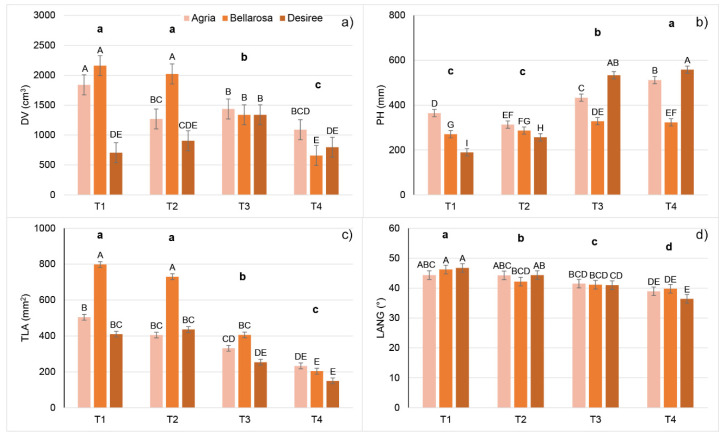
Mean values and double standard error of the mean for the morphological traits: (**a**) digital volume (DV), (**b**) plant height (PH), (**c**) total leaf area (TLA), and (**d**) leaf angle (LANG) of potato cultivars (Agria, Bellarosa, and Desiree) after ten days of growth in four different temperature treatments: 20/15 °C (T1), 25/20 °C (T2), 30/25 °C (T3), and 35/30 °C (T4). Different uppercase letters indicate significant differences for cultivar × treatment interaction; different lowercase letters in bold indicate significant differences between temperature treatments (Tukey’s HSD test *p* < 0.05).

**Figure 2 plants-11-03534-f002:**
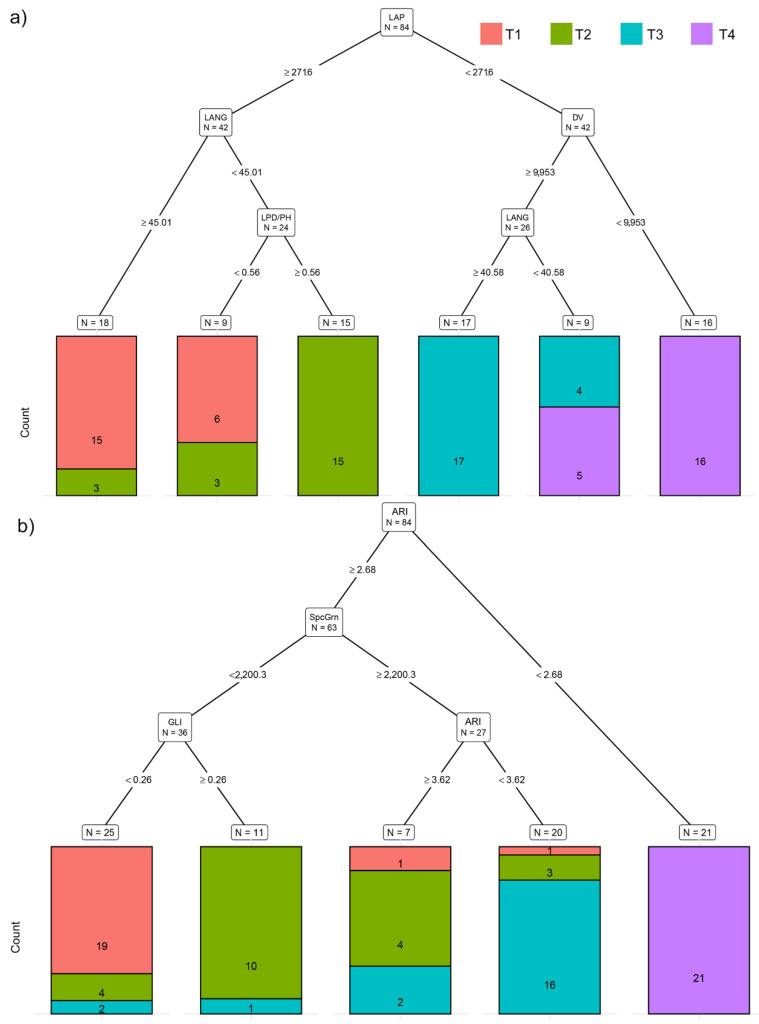
Visualization of classification tree for (**a**) morphological traits (MORPH) and (**b**) multispectral traits (MST). Each node shows the predicted class (Treatment). Below each node are counts of observations classified to each class (in the following order: T1, T2, T3, and T4) and the percentage of observations in the node. Each split shows the selected variable and cutoff value for each step in the classification tree. Legend: T1 (20/15 °C), T2 (25/20 °C), T3 (30/25 °C), and T4 (35/30 °C).

**Figure 3 plants-11-03534-f003:**
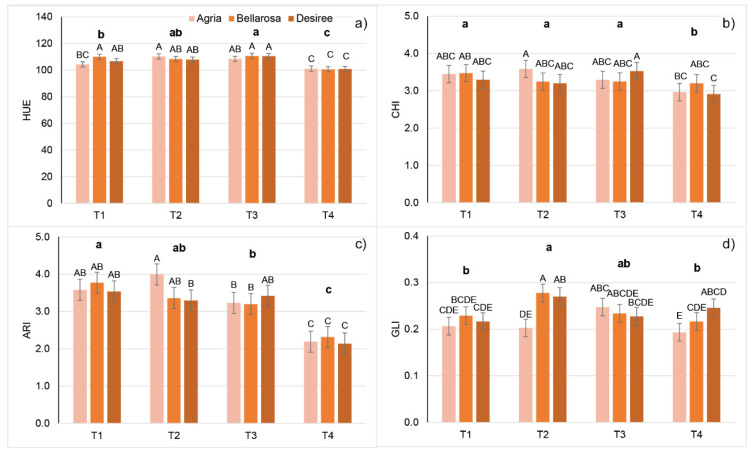
Mean values and double standard error of the mean for the multispectral traits: (**a**) hue (HUE), (**b**) chlorophyll index (CHI), (**c**) anthocyanin index (ARI), and (**d**) green leaf index (GLI) of potato cultivars (Agria, Bellarosa, and Desiree) after ten days of growth in four different temperature treatments: 20/15 °C (T1), 25/20 °C (T2), 30/25 °C (T3), and 35/30 °C (T4). Different uppercase letters indicate significant differences for cultivar × treatment interaction; different lowercase letters in bold indicate significant differences between temperature treatments (Tukey’s HSD test *p* < 0.05).

**Figure 4 plants-11-03534-f004:**
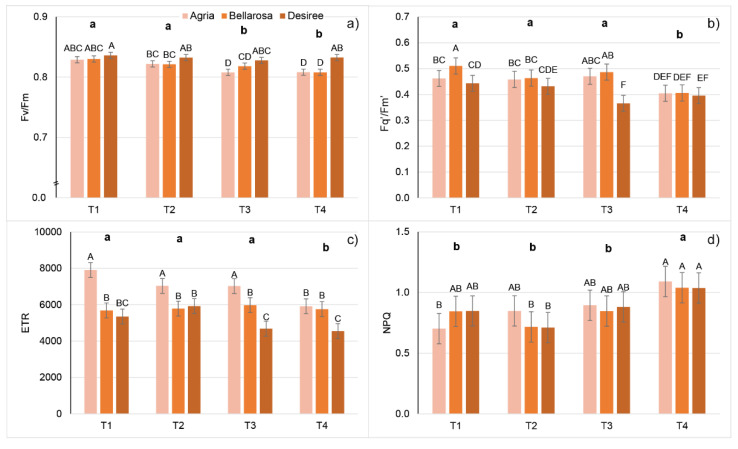
Mean values and double standard error of the mean for the chlorophyll fluorescence traits: (**a**) maximum quantum yield of PSII (F_v_/F_m_), (**b**) effective quantum yield of PSII (F_q_’/F_m_’), (**c**) non-photochemical quenching (NPQ), and (**d**) electron transport rate (ETR) of potato cultivars (Agria, Bellarosa, and Desiree) after ten days of growth in four different temperature treatments: 20/15 °C (T1), 25/20 °C (T2), 30/25 °C (T3), and 35/30 °C (T4). Different uppercase letters indicate significant differences for cultivar × treatment interaction; different lowercase letters in bold indicate significant differences between temperature treatments (Tukey’s HSD test *p* < 0.05).

**Figure 5 plants-11-03534-f005:**
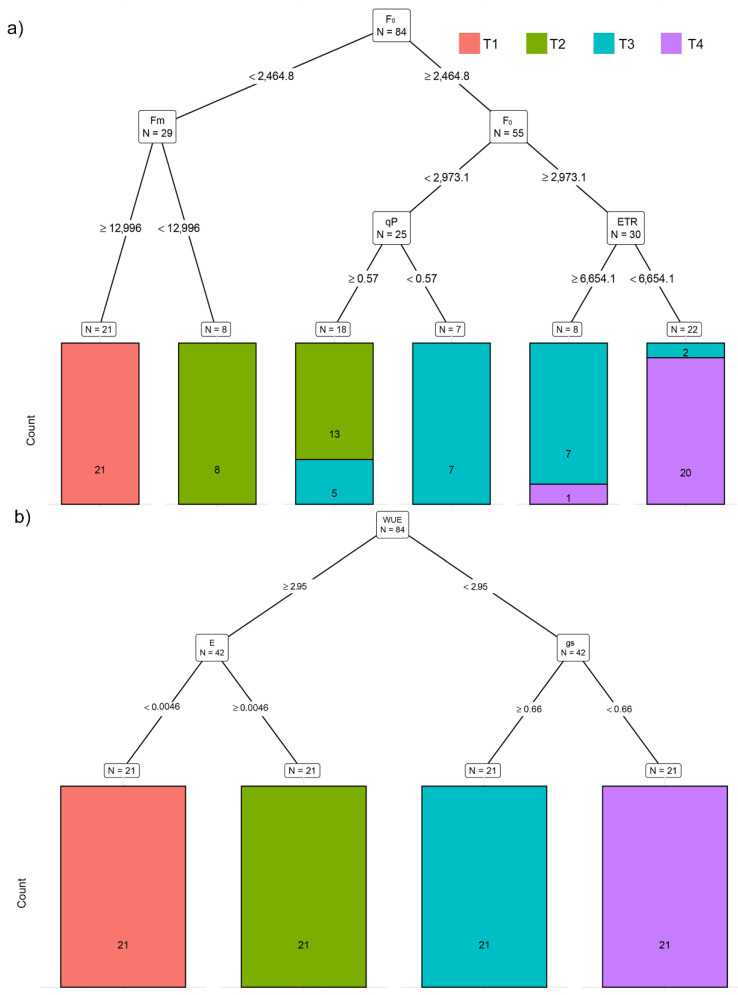
Visualization of classification tree for: (**a**) chlorophyll fluorescence traits (CFT) and (**b**) gas exchange traits (GEXT). Each node shows the predicted class (Treatment). Below each node are counts of observations classified into each class (in the following order: T1, T2, T3, and T4) and the percentage of observations in the node. Each split shows the selected variable and cutoff value for each step in the classification tree. Legend: T1 (20/15 °C), T2 (25/20 °C) T3 (30/25 °C), and T4 (35/30 °C).

**Figure 6 plants-11-03534-f006:**
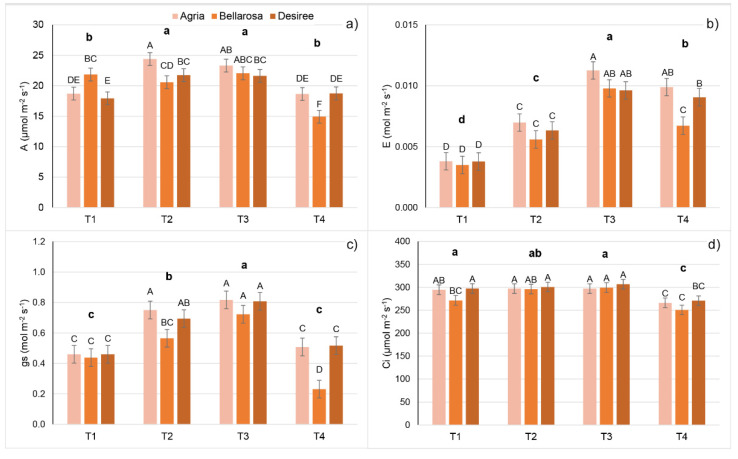
Mean values and double standard error of the mean for the gas exchange traits: (**a**) Photosynthesis rate (A, µmol CO_2_ m^−2^ s^−1^), (**b**) transpiration rate (E, mol H_2_O m^−2^ s^−1^), (**c**) stomatal conductance to water vapor (gs, mol H_2_O m^−2^ s^−1^), and (**d**) the intercellular CO_2_ concentration (Ci, µmol CO_2_ m^−2^ s^−1^) of potato cultivars (Agria, Bellarosa, and Desiree) after ten days of growth in four different temperature treatments: 20/15 °C (T1), 25/20 °C (T2), 30/25 °C (T3), and 35/30 °C (T4). Different uppercase letters indicate significant differences for cultivar × treatment interaction; different lowercase letters in bold indicate significant differences between temperature treatments (Tukey’s HSD test *p* < 0.05).

**Figure 7 plants-11-03534-f007:**
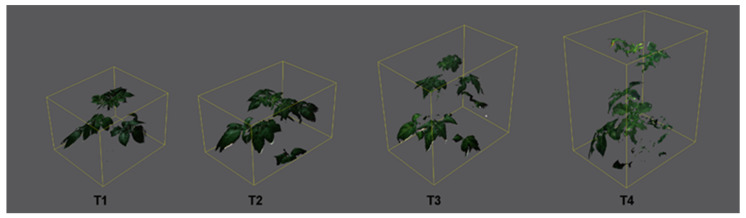
Color images of 3D plant models of potato plants scanned by PlantEye F500 grown for ten days in four different temperature treatments: 20/15 °C (T1), 25/20 °C (T2), 30/25 °C (T3), and 35/30 °C (T4).

**Figure 8 plants-11-03534-f008:**
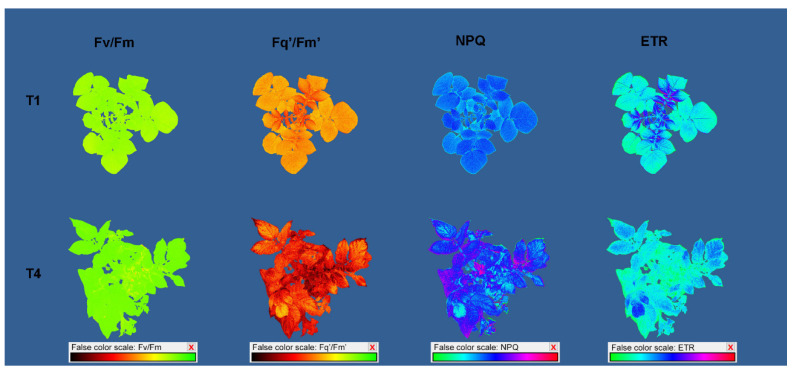
Potato pseudo-color images of the maximum quantum yield of PSII (F_v_/F_m_), the effective quantum yield of PSII (F_q_’/F_m_’), non-photochemical quenching (NPQ), and electron transport rate (ETR) captured by CropReporter after ten days of growth in different temperature treatments: 20/15 °C (T1) and 35/30 °C (T4).

**Figure 9 plants-11-03534-f009:**
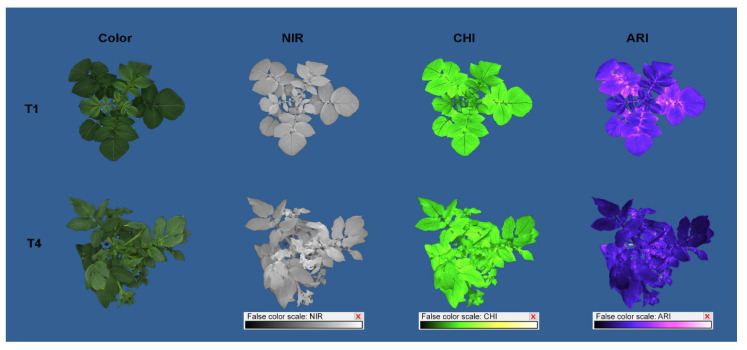
Potato color and pseudo-color images of reflectance in near-infrared (NIR), chlorophyll index (CHI), and anthocyanin index (ARI) captured by CropReporter after ten days of growth in different temperature treatments: 20/15 °C (T1) and 35/30 °C (T4).

## Data Availability

The data that support the findings of this study are available from the corresponding author [B.L.], upon reasonable request.

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
