# Peer review of "Study of High-Temperature-Induced Morphological and Physiological Changes in Potato Using Nondestructive Plant Phenotyping"

_plants, 2022, doi:10.3390/plants11243534_

Round 1

Reviewer 1 Report

Potato is one of the most important food crops, and high temperature is a critical unfavorable environmental condition reducing potato yields. Authors used nondestructive phenotyping techniques to study the effect of moderate heat stress on potato morphology and physiology, and found that leaf area projected (LAP), ARI, F0, and WUE are most responsive potato phenotypic traits to increased temperature. Overall, this manuscript is interesting and well written, and the results help provide marker traits for further studying potato responses to increased temperature. I have one minor concern for authors to consider improving the manuscript.

1. The statistical analysis is hard to understand in Fig. 1, 3, 4, 6. The comparison should be performed among four treatments (T1-T4) in each cultivar, not among three cultivars, because cultivars have different background. 

Author Response

Dear Reviewer, thank you for your suggestion.

Reviewer 1 suggestion:

  1. The statistical analysis is hard to understand in Fig. 1, 3, 4, 6. The comparison should be performed among four treatments (T1-T4) in each cultivar, not among three cultivars, because cultivars have different background. 

Answer:

We have changed the figures to accentuate differences among temperature treatments. Please see figures 1, 3, 4, and 6. However, the suggestions from other Reviewers who emphasize the importance of variability figures also show the interaction treatment x cultivar.

Reviewer 2 Report

The authors tested the applicability of new nondestructive phenotyping techniques by measuring different morphological and physiological parameters in correlation with the adaptation of potato, facing to the effects of climate change. Young potato plants were subjected to different heat stress regimes for ten days to select phenotypic traits most responsive to increased temperatures.

The analytical assessment of the study was adequate and the methods applied can be considered as up-to-date.

The manuscript involves valuable data about the morphological and physiological responses affected by heat stress. The main achievement of the project was that they have found the marker traits for further studying potato responses to increased temperatures.

However, it could be considered that young plants responded more intensively to the abiotic factors than the same varieties in cultivation after several months of adaptation to field contditions. Ontogenetic phases may also play important role in plant responses to the changing environment.

I suggest that the Discussion part should be shortened and prepared in a more concise way, while the Conclusions are to be extended by 1-2 sentences  about further prospects of the study, reflecting partly for the possible implementation of the results into the growing practice.

Author Response

Dear Reviewer, thank you for your suggestions.

Reviewer 2

  1. It could be considered that young plants responded more intensively to the abiotic factors than the same varieties in cultivation after several months of adaptation to field conditions. Ontogenetic phases may also play important role in plant responses to the changing environment.
  2. I suggest that the Discussion part should be shortened and prepared in a more concise way, while the Conclusions are to be extended by 1-2 sentences about further prospects of the study, reflecting partly for the possible implementation of the results into the growing practice.

Answer: 

We have reduced the discussion (please see deleted LINES: 262-266; 270-274; 278; 281; 288-292; 301-304; 309-312; 329; 336-344; 354; 366; 375-38)

We have included a discussion about the importance of confirmation results of this study under field conditions and also in different phenological phases. For this, we have used several new references (Please see the new references: 35, 36, 37. In addition, due to the suggestions from Reviewer 3 we have included the discussion about the possible effect of a higher number of cultivars and contrasting cultivars (heat tolerant s heat sensitive) on the results of this study. Please see LINES 382 – 401.

We have added sentences about the further prospect of the study and the possible application of selected traits in practice; please see LINES 546 – 554.

Reviewer 3 Report

In the present manuscript, authors used nondestructive phenotyping techniques such as multispectral and chlorophyll fluorescence imaging, 3D multispectral scanning, and gas exchange analysis to study potato plants' heat stress responses. In this study, three varieties of potato (Agria, Bellarosa and Desiree) were grown under four temperatures: 20/15°C (T1), 25/20°C (T2), 30/25°C (T3), and 35/30°C (T4). These heat-stressed plants were phenotyped and data were analyzed.

1.     Please add the graph legend to figure 4. I think it is T1, T2, T3 and T4.

2.     There is not much difference between the value of the three species used. To demonstrate the importance of the technique, the author should include some contrasting potato cultivars to illustrate the phenotypic differences.

Author Response

Dear Reviewer, thank you for your time and suggestions.

Reviewer 3 suggestions:

  1. Please add the graph legend to figure 4. I think it is T1, T2, T3 and T4.

Answer: Due to the suggestion from Reviewer 1 we have changed the figures: Fig 1, 3,4 and 6. The legend has been added to these new figures.

  1. There is not much difference between the value of the three species used. To demonstrate the importance of the technique, the author should include some contrasting potato cultivars to illustrate the phenotypic differences.

Answer: 

The idea of this study was to evaluate the effect of increased temperatures on potato morphology and physiology using nondestructive plant phenotyping techniques, not to compare phenotypic traits among different potato cultivars or to screen for heat-tolerant cultivars.

Because of it, we have deleted “different potato cultivars” from the aim; please see LINES 101-103.

In addition, although phenotypic differences among potato cultivars under heat stress is an important topic, this would be a different study which would have to include a higher number of cultivars. In this study, we included three commercially important potato cultivars. They are different in tuber skin colour, maturity shoot morphology etc. Our results also show a significant difference among cultivars and significant treatment x cultivar interaction on most studied traits (please see the ANOVA table in supplementary table S2). However, as you have stated, studied potato cultivars similarly responded to the increased temperatures. We have pointed this out in the discussion section. Please see LINES: 259-262.

Also, we agree that including very contrasting potato cultivars (for example, heat sensitive vs heat tolerant) could affect the accuracy of the recursive partitioning models and the selected traits for discrimination among different temperature treatments. Unfortunately, at this point, it is not feasible to repeat the whole experiment and include such cultivars in the study. However, to address this issue, we have extended the discussion. In addition, due to the suggestions from Reviewer 2 we have included the discussion about the importance of confirming the results of this study under the field conditions and also in different phenological phases.  Please see LINES 382 – 401. These things were also emphasized once more in the conclusions (please see LINES: 546-554).

Please see the attached corrected manuscript.

Round 2

Reviewer 3 Report

The manuscript can be accepted